# The α-7 Nicotinic Receptor Positive Allosteric Modulator Alleviates Lipopolysaccharide Induced Depressive-like Behavior by Regulating Microglial Function, Trophic Factor, and Chloride Transporters in Mice

**DOI:** 10.3390/brainsci14030290

**Published:** 2024-03-19

**Authors:** Sami Alzarea, Amna Khan, Patrick J. Ronan, Kabirullah Lutfy, Shafiqur Rahman

**Affiliations:** 1Department of Pharmaceutical Sciences, College of Pharmacy, South Dakota State University, Brookings, SD 57007, USA; 2Research Service, Sioux Falls VA Healthcare System, Sioux Falls, SD 57105, USA; 3Department of Psychiatry and Basic Biomedical Sciences, University of South Dakota Sanford School of Medicine, Sioux Falls, SD 57105, USA; 4College of Pharmacy, Western University of Health Sciences, Pomona, CA 91766, USA

**Keywords:** nicotinic receptor, major depressive disorder, neuroinflammation, microglia, α7 nicotinic receptor positive allosteric modulator, mice

## Abstract

Neuroinflammation contributes to the pathophysiology of major depressive disorder (MDD) by inducing neuronal excitability via dysregulation of microglial brain-derived neurotrophic factor (BDNF), Na-K-Cl cotransporter-1 (NKCC1), and K-Cl cotransporter-2 (KCC2) due to activation of BDNF-tropomyosin receptor kinase B (TrkB) signaling. Allosteric modulation of α7 nAChRs has not been investigated on BDNF, KCC2, and NKCC1 during LPS-induced depressive-like behavior. Therefore, we examined the effects of PNU120596, an α7 nAChR positive allosteric modulator, on the expression of BDNF, KCC2, and NKCC1 in the hippocampus and prefrontal cortex using Western blot analysis, immunofluorescence assay, and real-time polymerase chain reaction. The effects of ANA12, a TrkB receptor antagonist, on LPS-induced cognitive deficit and depressive-like behaviors were determined using the Y-maze, tail suspension test (TST), and forced swim test (FST). Pharmacological interactions between PNU120596 and ANA12 were also examined. Experiments were conducted in male C57BL/6J mice. LPS administration (1 mg/kg) resulted in increased expression of BDNF and the NKCC1/KCC2 ratio and decreased expression of KCC2 in the hippocampus and prefrontal cortex. PNU120596 pretreatment (4 mg/kg) attenuated the LPS-induced increase in the expression of BDNF and NKCC1/KCC2 ratio and the reduction in KCC2 expression in these brain regions. In addition, ANA12 (0.25 or 0.50 mg/kg) reduced the LPS-induced cognitive deficit and depressive-like behaviors measured by a reduced spontaneous alternation in the Y-maze and increased immobility duration in TST and FST. Coadministration of PNU120596 (1 mg/kg) and ANA12 (0.25 mg/kg) prevented the LPS-induced cognitive deficit and depressive-like behaviors. Overall, PNU120596 prevented the LPS-induced depressive-like behavior by likely decreasing neuronal excitability via targeting microglial α7 nAChR in the hippocampus and prefrontal cortex.

## 1. Introduction

Neuroinflammation has been recently proposed to be involved in the pathophysiology of major depressive disorder (MDD) [1,2]. This theory was realized because proinflammatory cytokines were found at high levels in depressed patients [3,4,5,6,7]. Moreover, cytokine immunotherapy can develop MDD-related symptoms in patients [8]. Healthy volunteers exhibit MDD-related symptoms when receiving agents activating the immune system, such as vaccines and endotoxins [9,10,11,12,13]. An immune challenge by lipopolysaccharide (LPS, endotoxin) leads to depressive-like behavior in rodents [14,15,16].

Microglia-neuron communication functions bidirectionally. Microglia impart significant influences on numerous aspects of neuronal functions [17,18]. Similarly, microglial brain-derived neurotrophic factor (BDNF) also induces neuronal excitability [19]. BDNF plays an important role in the regulation of neurogenesis and neurite outgrowth in a normal physiological condition [20]. Genetic studies have indicated that BDNF is implicated in depressive-like behavior in animals [21,22]. Similarly, it has also been shown to alter neuronal excitability via regulation of γ-aminobutyric acid (GABA) release from interneurons [23]. In addition, BDNF could be regulated during inflammation [24,25]. Interestingly, the binding of BDNF to its receptor tyrosine kinase B (TrkB) generates the signal that dysregulates transmission of GABA. The dysregulation of GABAergic transmission is associated with K^+^–Cl^−^ cotransporter (KCC2) downregulation and Na^+^–K^+^–Cl^−^ cotransporter (NKCC1) upregulation [26,27,28].

Dysfunctions of NKCC1 are considered to be associated with various neuropsychiatric disorders, such as depression, autism, and schizophrenia [29,30]. Recent research has shown that increased secretion of interleukin-18 (IL-18) mediates depressive-like behaviors via promoting the IL-18/NKCC1 signaling pathway [29]. Moreover, altered NKCC1/KCC2 expression has been observed in the brains of depressive and schizophrenic patients [31,32]. Furthermore, investigation of postmortem brain tissues has identified decreased expression of KCC2 transcripts in adults diagnosed with schizophrenia and affective mood disorder [32].

Nicotinic acetylcholine receptors (nAChRs) have been targeted for drug development in cognitive and neurodegenerative disorders in preclinical studies [33,34]. The homomeric α7 nAChR subtype is widely expressed by neuronal and non-neuronal cells, such as microglia, in the central nervous system (CNS) [35,36]. Microglia have been suggested to regulate immune responses via the anti-inflammatory cholinergic pathway, involving interaction between the nervous and immune systems [36]. The α7 nAChRs are included in the modulation of microglial activation, the primary source of neuroinflammation in the brain [37,38,39]. Moreover, these receptors are widely expressed in MDD-relevant brain regions, such as the hippocampus and prefrontal cortex, implicated in the regulation of emotion and behavior [35,40,41]. Therefore, the anti-inflammatory role of α7 nAChRs could be a promising target to treat disorders associated with inflammation [36,42]. Microglial activation causes the release of BDNF as an inflammatory mediator and is involved in the dysregulation of chloride ion concentration, which is regulated by GABAergic neurotransmission via NKCC1 upregulation and KCC2 downregulation. Interruption of intracellular chloride ion concentration subsequently induces neuronal excitability. Thus, targeting BDNF, NKCC1, and KCC2 within the hippocampus and prefrontal cortex via α7 nAChR PAM might have potential therapeutic utility for MDD.

Ligands acting on the orthosteric binding site of α7 nAChR may allow rapid desensitization that limits the receptor function [43,44]. Therefore, modulation of the α7 nAChR by ligands acting on the allosteric binding site represents an alternative aspect of overcoming desensitization. Evidence indicates that α7 nAChR positive allosteric modulators (PAMs) type II prevent desensitization and enhance cholinergic neurotransmission [43,44]. As a result, the α7 nAChR PAMs could be an appropriate choice to ensure the maximum anti-inflammatory properties of the α7 nAChRs. The α7 nAChR PAM, PNU120596, has been found to prevent LPS-induced depressive-like behavior in mice [37]. However, the impact of allosteric modulation of microglial α7 nAChR on the regulation of BDNF, NKCC1, and KCC2 associated with MDD remains unknown.

The objective of our present study was to determine the effects of PNU120596, an α7 nAChR PAM, on BDNF, NKCC1, and KCC2 expression in the hippocampus and prefrontal cortex in the LPS-induced mouse model of MDD. We also examined the pharmacological interaction between PNU120596 and ANA12, an antagonist of TrkB, on LPS-induced depressive-like behaviors.

## 2. Materials and Methods

### 2.1. Animals

Male C57BL/6J mice (weighing 20–30 g) were purchased from Jackson Laboratory (Bar Harbor, ME, USA). Mice were housed in groups of 5 in standard shoebox cages (29 × 18 × 12 cm) under standard laboratory conditions (22 ± 2 °C, relative humidity 50–60%) and maintained on a 12 h light/dark cycle (lights on at 0600 h) with free access to food and water. The behavioral experiments were conducted between 0900 and 1600 h, and mice (10–12 weeks of age) were allowed to habituate to the testing room with dim light for at least 1 h. A counter-balanced design was used to control any order effects. All procedures described herein follow the National Institutes of Health Guide for the Care and Use of Laboratory Animals and were approved by the Institutional Animal Care and Use Committee at South Dakota State University. Good Laboratory Practice and ARRIVE guidelines were followed. All efforts were made to ensure minimal animal suffering.

### 2.2. Drugs and Treatment

LPS (from Escherichia coli, serotype 0127:B8) and methyllycaconitine (MLA), an α7 nAChR antagonist, were purchased from Sigma-Aldrich (St. Louis, MO, USA) and dissolved in normal saline (0.9% NaCl). PNU120596 was purchased from Tocris Bioscience (Ellisville, MO, USA) and reconstituted in normal saline with 5% dimethyl sulfoxide (DMSO) and 5% Solutol (Sigma, St. Louis, MO, USA). ANA-12 was purchased from MedChem Express (Monmouth Junction, NJ, USA) and reconstituted in normal saline with 1% DMSO and 0.5% tween 80. All chemicals were administered intraperitoneally in a volume of 10 mL/kg of body weight.

### 2.3. Experimental Procedure

Mice were treated with PNU120596 (1 or 4 mg/kg) followed by LPS (1 mg/kg) administration after 0.5 h, as previously described [37]. MLA (3 mg/kg) was injected 10 min before PNU120596 (4 mg/kg) administration. Brain tissues were collected for analyses 24 h after LPS administration because significant depressive-like behavior appears at this time and is prevented by PNU120596 effects [16,37]. The hippocampi and prefrontal cortices were dissected, frozen on dry ice, and stored at −80 °C until further analysis. For behavioral studies, mice were treated with ANA12 (0.25 or 0.50 mg/kg) 23 h after LPS administration to determine the antidepressant-like effects of ANA12. The pharmacological interaction between PNU120596 and ANA12 on LPS-induced depressive-like behavior was tested using locomotor activity(LMA), Y-maze, tail suspension test (TST), and forced swim test(FST).

### 2.4. Western Blot Analysis

Western blot analysis was carried out as described previously, with minor modifications [45]. Briefly, brain tissue samples were homogenized in modified RIPA buffer containing Dulbecco’s phosphate-buffered saline (pH 7.4), 0.1% sodium dodecyl sulfate, 1% IGEPAL CA-630, and a protease inhibitor. Each sample was centrifuged (14,000× *g*, 20 min at 4 °C), and the supernatant was collected. Total protein concentration in the brain samples was determined by a bicinchoninic acid assay (Pierce, Rockford, IL, USA). Equal amounts of protein (50 μg) were loaded onto gels for sodium dodecyl sulfate (SDS) polyacrylamide gel electrophoresis. Separated proteins were transferred onto a PVDF membrane (MilliporeSigma, Billerica, MA, USA). Membranes were then blocked on a gyro-rocker with 5% nonfat dry milk in Tris-buffered saline/0.1% tween-20 and subsequently incubated overnight at 4 °C with primary antibodies for BDNF (1:300, rabbit polyclonal, Santa Cruz Biotech, Iowa, IA, USA), p-CREB (1:500, rabbit polyclonal, Santa Cruz Biotech, USA), or β-tubulin (E7-S, 1:5000, mouse monoclonal, University of Iowa, Iowa, IA, USA). After incubation, membranes were incubated with appropriate horseradish peroxide-conjugated secondary antibodies. Bound antibodies were detected with enhanced chemiluminescence prime reagent (Amersham, Buckinghamshire, UK), and protein quantification was performed using densitometric analysis.

### 2.5. Immunofluorescence Assay

Immunofluorescence assays were performed as described previously [37]. Briefly, 14 µM coronal tissue slices were washed with phosphate-buffered saline (PBS). The tissues were retrieved using 0.01 M citrate buffer (pH 6.0) heated at 90 °C in a water bath for 10 min. Brain tissue sections for the hippocampus and prefrontal cortex were then blocked and incubated with a primary antibody against BDNF (1:50, Santa Cruz Biotech, USA) overnight at 4 °C. Tissue sections were then incubated with a secondary antibody labeled with fluorescein isothiocyanate (FITC) (Santa Cruz Biotechnology, Dallas, TX, USA). The slides were mounted with mounting medium containing 4′,6′-diamidino-2-phenylindole (DAPI) for nuclear staining and prolong anti-fade reagent (Santa Cruz Biotechnology, Dallas, TX, USA). Immunofluorescence was examined using a laser-scanning confocal microscope (Fluoview FV1200, Olympus, Tokyo, Japan). The integrated density of protein immunoreactivity was performed using Image J software (v1.8.0, NIH, Bethesda, MD, USA).

### 2.6. Quantitative Real-Time Polymerase Chain Reaction

Total RNA was isolated from brain tissue using TRIzol reagent (Invitrogen, Carlsbad, CA, USA) according to the manufacturer’s instructions. One microgram of total RNA was reverse transcribed using a reverse transcription system to make cDNA using the High-Capacity cDNA Reverse Transcription Kit (Applied Biosystems, Carlsbad, CA, USA) and Master Cycler Personal (Eppendorf, Hauooauge, NY, USA) as described previously [38]. Approximately 20 µL of cDNA was yielded, out of which 1 μL was used for each real-time PCR analysis. Primers sequences were obtained from Integrated DNA Technologies (Coralville, Iowa, USA) and are displayed in Table 1. The Ct value was measured for each gene, and the relative expression of each gene was calculated using the delta-delta Ct method.

### 2.7. Behavioral Tests

#### 2.7.1. Locomotor Activity

The locomotor activity was assessed as described previously [45]. Briefly, the animals were individually placed in the periphery of the cage (40 cm × 40 cm × 35 cm). The mice were allowed to explore the chamber for 15 min, with the first 5 min being the acclimatization period and the last 10 min being the test period. A camera was vertically held 100 cm above the cage to record the animals’ behavior. Then, the videos were analyzed with ANY-maze software (v7.37, Stoelting Co., Wood Dale, IL, USA) to measure the total distance traveled (m) by each animal.

#### 2.7.2. Y-Maze

The Y-maze was performed as described previously [15]. Briefly, the Y-maze apparatus was made of gray Plexiglas and consisted of 3 closed arms (35 cm × 5 cm × 10 cm) at a 120° angle from each other. Each mouse was placed at the center of the maze and allowed to explore freely for 8 min. During the test, each mouse was recorded, and the videos were analyzed to measure spontaneous alternations. The alternations were counted when the 3 different arms were visited by the mouse without making a return entry to an already visited arm. Percent spontaneous alternations (alternate arm entries/total number of entries) × 100 was also calculated. An arm entry was defined as the presence of all 4 feet of the animal in 1 arm. The apparatus was thoroughly cleaned with 70% ethanol after the removal of each mouse.

#### 2.7.3. Tail Suspension Test

The TST was performed as described previously [46]. Briefly, each mouse was suspended in the tail suspension test chamber with a distance of around 45 cm from the floor by attaching the tail to a hook in the chamber with a distance from the tip of the tail of around 1 cm by using adhesive tape. The test was conducted for 6 min by recording each mouse. The videos were analyzed to measure immobility time for each mouse by observing any absence of leg or body movements as immobility.

#### 2.7.4. Forced Swim Test

The FST was performed as described previously [47]. Briefly, mice were placed individually in a cylindrical Plexiglas^®^ tank (45 cm high × 20 cm diameter), which was filled with 25 cm of water (25 ± 1 °C), and allowed to swim for 6 min. During the test, each mouse was recorded, and the videos were analyzed to measure immobility time. Immobility was counted when no additional activities were observed other than those required to keep the head above water. Mice were removed from the cylinder immediately after the test, dried with paper towels, and returned to their home cages.

### 2.8. Statistical Analysis

Biochemical data were analyzed using one-way ANOVA. Two-way ANOVA was conducted (LPS vs. control × treatments) for each behavioral test. Tukey’s post hoc test was performed for multiple comparisons using GraphPad Prism (GraphPad Inc., San Diego, CA, USA). Results were expressed as mean ± S.E.M. The difference between treatments was considered significant at *p* < 0.05.

## 3. Results

### 3.1. Effects of PNU120596 on the Expression of BDNF and p-CREB in the DG and CA1 Regions of the Hippocampus and Medial Prefrontal Cortex

We examined the effects of PNU120596 on LPS-induced increases in BDNF expression in the DG, CA1, and medial prefrontal cortex (Figure 1A). One-way ANOVA indicated that PNU120596 pretreatment had significant effects on LPS-elevated BDNF expression in the DG and CA1 regions of the hippocampus (F_4,19_ = 7.763; *p* < 0.001) and medial part of the prefrontal cortex (F_4,20_ = 7.272; *p* < 0.001). The post hoc test for multiple comparisons revealed that LPS (1 mg/kg) significantly (*p* < 0.01) increased BDNF expression in the DG and CA1 regions of the hippocampus and medial prefrontal cortex compared to control. In addition, PNU120596 (4 mg/kg) significantly (*p* < 0.05) decreased BDNF expression in the DG and CA1 regions of the hippocampus and medial prefrontal cortex compared to the LPS-treated group. In contrast, MLA significantly blocked PNU120596′s effects in all these regions. To assess whether p-CREB expression is modified during BDNF changes associated with neuroinflammation, we evaluated the effects of PNU120596 on p-CREB expression in the DG and CA1 regions of the hippocampus and medial prefrontal cortex (Figure 1B). One-way ANOVA indicated that LPS and PNU120596 had no significant effects on p-CREB expression in the DG and CA1 regions of the hippocampus (F_4,17_ = 0.4929; *p* = 0.7411) and medial part of the prefrontal cortex (F_4,17_ = 0.1825; *p* = 0.9443).

### 3.2. Effects of PNU120596 on BDNF Immunoreactivity in the DG and CA1 Regions of the Hippocampus and Medial Prefrontal Cortex

To evaluate the effects of PNU120596 on LPS-induced increases in BDNF expression, BDNF immunoreactivity was tested in the DG, CA1, and medial prefrontal cortex (Figure 2). One-way ANOVA showed that PNU120596 significantly decreased LPS-induced BDNF expression in DG (F_2,15_ = 6.523; *p* < 0.01), CA1 (F_2,15_ = 6.115; *p* < 0.05), and medial prefrontal cortex (F_2,9_ = 6.652; *p* < 0.05). The post hoc test for multiple comparisons revealed that LPS (1 mg/kg) significantly increased BDNF expression compared to control. Furthermore, PNU120596 (4 mg/kg) significantly (*p* < 0.05) reduced BDNF expression in all these regions compared to the LPS treated group.

### 3.3. Effects of PNU120596 on the mRNA Expression of KCC2 and the NKCC1/KCC2 Ratio in the Hippocampus and Prefrontal Cortex

To examine the effects of PNU120596 on neuronal excitability in the hippocampus and prefrontal cortex, we quantified the mRNA expression of KCC2 and the NKCC1/KCC2 ratio (Figure 3). One-way ANOVA indicated that PNU120596 pretreatment restored LPS-mediated reduction in KCC2 expression (Figure 3A) in the hippocampus (F_4,24_ = 4.359; *p* < 0.01) and prefrontal cortex (F_4,25_ = 5.092; *p* < 0.01), and the LPS-induced NKCC1/KCC2 expression (Figure 3B) in the hippocampus (F_4,21_ = 7.201; *p* < 0.001) and prefrontal cortex (F_4,20_ = 5.551; *p* < 0.01). The post hoc test for multiple comparisons revealed that LPS (1 mg/kg) clearly decreased mRNA of KCC2 and increased mRNA of the NKCC1/KCC2 ratio in the hippocampus and prefrontal cortex compared to control. In addition, PNU120596 (4 mg/kg) significantly blocked the LPS-induced reduction in mRNA of KCC2 and decreased the LPS-elevated mRNA of the NKCC1/KCC2 ratio in the hippocampus and prefrontal cortex. In contrast, MLA significantly blocked PNU120596′s effects on the expression of KCC2 and the NKCC1/KCC2 ratio in both brain regions.

### 3.4. Effects of ANA12 on LPS-Induced Depressive-like Behavior

To determine whether the blockade of TrkB can reduce LPS-induced cognitive deficit and depressive-like behavior, we examined the effects of ANA12 on LMA, Y-maze, TST, and FST. The effects of ANA12 (0.25 or 0.50 mg/kg) on the total distance traveled by the mice for locomotor activity 24 h after LPS injection (Figure 4B) were analyzed. A two-way ANOVA indicated that the locomotor activity was not significantly different between the treatment groups (F_2,32_ = 0.05983; *p* = 0.9420). The effects of ANA12 on spontaneous alternations for cognitive deficit-like behavior in the Y-maze 25.5 h after LPS injection are shown in Figure 4C. A two-way ANOVA indicated that ANA12 significantly (F_2,33_ = 3.464; *p* < 0.05) reversed the LPS-induced reduction in spontaneous alternations. The effects of ANA12 on immobility time for depressive-like behavior in TST and FST, 27 and 28.5 h, respectively after LPS injection are shown in Figure 4D,E. A two-way ANOVA indicated that ANA12 significantly reversed LPS-induced increases in immobility time in TST (F_2,33_ = 3.428; *p* < 0.05) and FST (F_2,33_ = 4.486; *p* < 0.05).

### 3.5. Combination Effects of PNU120596 and ANA12 on LPS-Induced Depressive-like Behavior

To evaluate whether there is an interaction between α7 nAChR and the TrkB receptor, we determined if subthreshold doses of PNU120596 and ANA12 could produce antidepressant and pro-cognitive-like effects in LMA, Y-maze, TST, and FST in an inflammatory mouse model of depression. The combination effects of PNU120596 (1 mg/kg) and ANA12 (0.25 mg/kg) on locomotor activity, measured as the total distance traveled by mice 24 h after LPS injection, are shown in Figure 5B. A two-way ANOVA indicated that the locomotor activity was not significantly different between the treatment groups (F_2,32_ = 0.09898; *p* = 0.9060). In addition, the combination effects of PNU120596 and ANA12 on spontaneous alternations for cognitive deficit-like behavior in Y-maze 25.5 h after LPS injection are shown in Figure 5C. A two-way ANOVA revealed that PNU120596 and ANA12 coadministration significantly (F_2,33_ = 3.789; *p* < 0.05) prevented LPS-induced reduction in spontaneous alternations. Furthermore, the combination effects of PNU120596 and ANA12 on immobility time for depressive-like behavior in TST and FST at 27 and 28.5 h, respectively after LPS injection are shown in Figure 5D,E. A two-way ANOVA revealed that PNU120596 and ANA12 coadministration significantly prevented the LPS-induced increase in immobility time in TST (F_2,33_ = 3.547; *p* < 0.05) and FST (F_2,37_ = 3.506; *p* < 0.05).

## 4. Discussion

The present findings indicate that systemic inflammation increases the expression of BDNF in the hippocampus and prefrontal cortex 24 h after LPS administration. These effects are associated with a reduction in mRNA expression of KCC2 and an increase in mRNA expression of the NKCC1/KCC2 ratio. Positive allosteric modulation of α7 nAChR via PNU120596 attenuates the changes induced by LPS in the expression of BDNF, KCC2, and the NKCC1/KCC2 ratio in the hippocampus and prefrontal cortex. These effects of PNU120596 are blocked by the action of MLA, an α7 nAChR antagonist. Interestingly, the treatments had no effects on p-CREB expression in the hippocampus or prefrontal cortex. Blockade of BDNF receptor TrkB by ANA12 reduces LPS-induced cognitive deficits and depressive-like behaviors. Coadministration of subthreshold doses of PNU120596 and ANA12 prevents LPS-induced cognitive deficits and depressive-like behaviors.

Innate immune system activation by systemic administration of LPS is associated with the induction of neuroinflammation in the brain [2]. Neuroinflammation produces molecular and behavioral changes that lead to depressive-like behavior 24 h after LPS injection [2,16]. Our data demonstrate that a systemic LPS challenge promoted upregulation of BDNF and the NKCC1/KCC2 ratio and downregulation of KCC2 in the hippocampus and prefrontal cortex. These changes are associated with LPS-induced cognitive deficits and depressive-like behaviors that appeared significantly 24 h after LPS administration.

BDNF has been found to be dysregulated during neuroinflammation [48]. It is involved in the nuclear factor-κB (NF-κB) inflammatory signaling pathway [24,25]. Upregulation of BDNF is associated with the activation of NF-κB and the upregulation of proinflammatory cytokines [49]. BDNF is likely an inflammatory mediator involved in the LPS-induced neuroinflammation in the present study. Therefore, BDNF might be involved in the modulation of depressive-like behavior associated with neuroinflammation. Indeed, BDNF has been implicated in the modulation of depressive-like behavior in rodents. Along these lines, though somewhat controversial, it is noteworthy to mention that BDNF in the ventral tegmental area-nucleus accumbens (VTA-NAc) pathway is involved in the development of depression-like behavior [50,51]. For example, mice experiencing social defeat stress exhibit depressive-like behavior associated with central BDNF upregulation in the mesolimbic dopamine pathway [50,52]. Consistent with a previous report, upregulation of BDNF in the hippocampus is associated with LPS-induced depressive-like behavior [53]. Mice reared in communal nests show upregulation of hippocampal BDNF expression and depressive-like behavior [54]. Mice genetically modified to overexpress BDNF show anxiety-like behavior when exposed to chronic immobilization stress [55]. In an inflammatory mouse model of depression, anxiety-like behavior is comorbid with depressive-like behavior [15] as an indication of BDNF involvement in depressive-like behavior. However, viral-mediated suppression of the BDNF receptor TrkB in the VTA and NAc exhibits antidepressant-like behavior [51]. BDNF knockout mice exhibit antidepressant-like behavior in FST [21], suggesting the regulatory function of BDNF in depressive-like behavior. Antidepressants such as imipramine and fluoxetine reduce BNDF expression in the rodent hippocampus [53,56]. Based on all the studies, it is important to note that BDNF expression is distinct in specific brain regions. BDNF generally appears to have antidepressant effects in the hippocampus and prefrontal cortex [52,57]. In contrast, BDNF exhibits pro-depressive effects in the VTA-NAc pathway [50,51,52]. Thus, BDNF overexpression in distinct brain regions has been attributed to a high degree of variability in stress models, microglial activators, and the time of measuring these trophic responses.

Recently, we have shown that positive allosteric modulation of α7 nAChRs by PNU120596 regulates LPS-induced depressive-like behavior by decreasing microglial activation [37]. In addition, chronic stress contributes to the generation of neuroactive kynurenine metabolites, which leads to subsequent depressive-like behavior [58]. Recently, we have found that α7 nAChRs PAM prevents neurotoxic metabolite production and its release from the microglia in an inflammatory mouse model of MDD [59]. Additional evidence demonstrates that activation of α7 nAChRs results in an anti-inflammatory action [36,60,61], which could be the underlying mechanism of the antidepressant-like effects of PNU120596. However, the effect of PNU120596 on BDNF dysregulation associated with neuroinflammation is unknown. Our findings demonstrate that PNU120596 reduces LPS-induced upregulation of BDNF expression in the hippocampus and prefrontal cortex. These results depend on the activation of α7 nAChRs because MLA, an α7 nAChR antagonist, blocks PNU120596′s effects on BDNF expression.

Dysregulation of BDNF was suggested to cause activation of BDNF-TrkB signaling cascades that dysregulate NKCC1 and KCC2. These effects are associated with a decreased level of chloride ions that are regulated by GABAergic neurotransmission [62]. The impairment of GABAergic neurotransmission has been proposed to be involved in the pathophysiology of MDD and other inflammatory conditions [63]. Our prior research supports the idea that positive allosteric modulation of α7 nAChRs regulates inflammatory pain-like symptoms, likely by reducing neuronal excitability [64]. Previous evidence confirms that mice exposed to social defeat stress exhibit depressive-like behavior associated with KCC2 downregulation that leads to neuronal depolarization [65]. Pharmacological blockade of NKCC1, reducing neuronal chloride concentration, produces antidepressant-like effects in mice [66]. In the present study, administration of LPS leads to downregulation of KCC2 and upregulation of the NKCC1/KCC2 ratio in the hippocampus and prefrontal cortex. This dysregulation might be correlated to the development of depressive-like behavior associated with inflammation via the BDNF–TrkB signaling cascade.

PNU120596 attenuates LPS-induced downregulation of KCC2 and upregulation of the NKCC1/KCC2 ratio in the hippocampus and prefrontal cortex. MLA prevents the effects of PNU120596 from normalizing these factors dysregulated by LPS, suggesting the involvement of α7 nAChRs in the regulation of NKCC1 and KCC2 in these brain regions. These effects of PNU120596 could be another reason for its attenuation of LPS-induced depressive-like behavior, as shown previously [37].

Microglia, or innate immune cells, are the primary source of inflammation in the brain [67]. Previous studies have shown that activated microglia produce and release BDNF [23,68,69]. Microglial BDNF has been shown to phosphorylate the TrkB receptor as an indication for activation [70], which may promote signals to dysregulate NKCC1 and KCC2 [32]. During inflammation, NF-κB, a transcription factor, is activated to synthesize several inflammatory genes, including BDNF [25]. To differentiate between neuronal and microglial BDNF, we determined the expression of CREB, which is a transcription factor of BDNF in the neurons [71]. Interestingly, our results indicate that CREB expression does not change during inflammation. Increased expression of BDNF with LPS may not be produced by the neurons in the present study. Future studies are needed to determine whether microglial BDNF modulates neuron-microglia interactions in other brain regions. The heterogeneity of microglial BDNF function should also be taken into consideration when investigating such mechanisms [72].

PNU120596 has been shown to prevent LPS-induced microglial activation in the hippocampus and prefrontal cortex [37]. PNU120596 might promote anti-inflammatory signals to prevent the dysregulation of BNDF, NKCC1, and KCC2 associated with microglial activation. These anti-inflammatory pathways are independent of calcium ion influx during activation of microglial α7 nAChRs. Therefore, microglial α7 nAChRs have been proposed to be metabotropic receptors [60,73,74].

Given the increased expression of BDNF in the hippocampus and prefrontal cortex as a response to LPS treatment, activation of the TrkB receptor by BDNF is speculated to generate signals that dysregulate NKCC1 and KCC2 and consequently mediate LPS-induced depressive-like behavior. We used the TrkB receptor antagonist ANA12 to examine whether stimulation of the BDNF-TrkB cascade as a response to LPS treatment is adequate to promote depressive-like behavior. We found that ANA12 was able to decrease LPS-induced depressive-like behavior.

Previous studies have shown that ANA12 reduces depressive-like behavior in mice. The antidepressant-like effects of ANA12 have been suggested to be due to blockade of TrkB receptors in the NAc [75,76]. Our results did not show antidepressant-like effects of ANA12 at a basal level, whereas the antidepressant-like effects of ANA12 were noticeable during inflammation. These effects suggest that ANA12 likely prevents upregulated BDNF from activating its TrkB receptor. Therefore, the antidepressant-like effects of ANA12 are suggested to be specific to blockade of the TrkB receptor and its activation of the signal, which dysregulate NKCC1 and KCC2, as a response to systemic LPS challenge.

In this study, ANA12 exhibits pro-cognitive-like effects in an inflammatory mouse model of depression. Cognitive impairment was targeted in this study because cognitive impairment is one of the depression-related symptoms [77]. The hippocampus [15,78] and prefrontal cortex [14,79] have been demonstrated to regulate spatial memory and reversal learning, respectively. Therefore, molecular changes in these brain regions may reflect changes in cognitive deficit-like behavior. ANA12 prevents LPS-induced cognitive deficit-like behavior, probably because ANA12 may prevent dysregulated BDNF from activating TrkB signals interrupting NKCC1 and KCC2 in the hippocampus and prefrontal cortex. Moreover, PNU120596 prevented LPS-induced depressive-like behavior [37]. We have found that coadministration of subthreshold doses of PNU120596 and ANA12 reduced LPS-induced cognitive deficits and depressive-like behaviors. These data might indicate the interaction between α7 nAChR and the TrkB receptor by generating a partial anti-inflammatory signal from α7 nAChR and a partial blockade of the TrkB cascade induced by BDNF.

Our results did not show different effects on locomotor activity. Our previous study showed that locomotor activity decreases significantly 6 h after LPS administration as an indication of sickness behavior because of molecular changes associated with inflammation. The locomotor activity becomes normal 24 h after LPS injection, which is associated with the significant appearance of depressive-like behavior, suggesting a behavioral transition [16].

## 5. Conclusions

In these studies, we have shown that the α7 nAChR PAM, PNU120596, through allosteric modulation, reduces LPS-induced increases in the expression of BDNF and the NKCC1/KCC2 ratio and KCC2 downregulation in the hippocampus and prefrontal cortex during neuroinflammation. The coadministration of PNU120596 and ANA12 induced an additive/synergistic antidepressant effect, representing a pharmacological interaction for preventing LPS-induced depressive-like behavior. These effects are likely due to the anti-inflammatory effects of PNU120596 and/or antagonism of the TrkB signal by ANA12 (Figure 6). Therefore, the present study provides evidence for novel molecular mechanisms for the antidepressant-like effects of PNU120596 involving microglial α7 nAChR.

## Figures and Tables

**Figure 1 brainsci-14-00290-f001:**
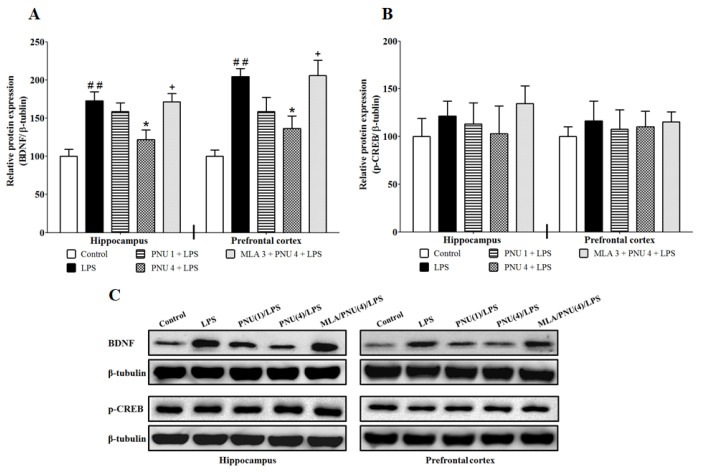
(**A**) Effects of PNU120596 (PNU) on LPS-induced increases in BDNF expression in the DG, CA1, and medial prefrontal cortex of mice. LPS (1 mg/kg) significantly (*p* < 0.01) increased BDNF expression in the DG, CA1, and medial prefrontal cortex compared to the control. PNU (4 mg/kg) significantly (*p* < 0.05) decreased BDNF expression in the DG, CA1, and medial prefrontal cortex compared to the LPS-treated group, and these changes were blocked by MLA. (**B**) Effects of PNU on p-CREB expression in the DG, CA1, and medial prefrontal cortex of mice. The treatments did not have significant effects on p-CREB expression. (**C**) Representative Western blots for BDNF and p-CREB expression from the DG, CA1, and medial prefrontal cortex. ^##^ *p* < 0.01, LPS (1 mg/kg) vs. control; * *p* < 0.05, PNU (4 mg/kg) plus LPS vs. LPS alone; **^+^** *p* < 0.05, MLA (3 mg/kg) plus PNU (4 mg/kg) plus LPS vs. PNU (4 mg/kg) plus LPS. Data are expressed as the mean ± S.E.M. of 4–6 mice/group.

**Figure 2 brainsci-14-00290-f002:**
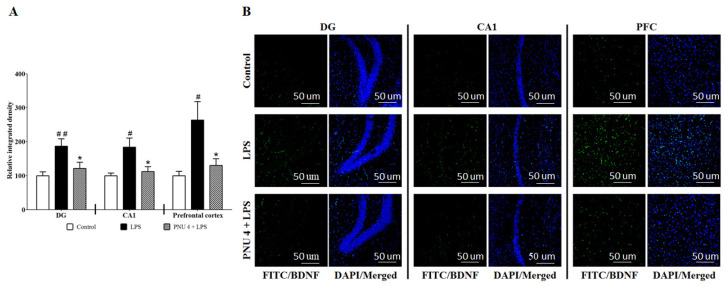
(**A**) Effects of PNU120596 (PNU) on immunoreactivity of BDNF in DG, CA1, and medial prefrontal cortex of mice. LPS (1 mg/kg) significantly increased BDNF expression in DG (*p* < 0.01), CA1 (*p* < 0.05), and medial prefrontal cortex (*p* < 0.05) compared to control. PNU (4 mg/kg) significantly (*p* < 0.05) decreased BDNF expression in the DG, CA1, and medial prefrontal cortex compared to the LPS-treated group. (**B**) Representative images of immunofluorescence in DG, CA1, and medial prefrontal cortex (PFC). Magnification: 20 X, scale bar = 50 μm. ^#^ *p* < 0.05 or ^##^ *p* < 0.01, LPS (1 mg/kg) vs. control; * *p* < 0.05, PNU (4 mg/kg) plus LPS vs. LPS alone. Data are expressed as the mean ± S.E.M. of *n* = 4–6 mice/group.

**Figure 3 brainsci-14-00290-f003:**
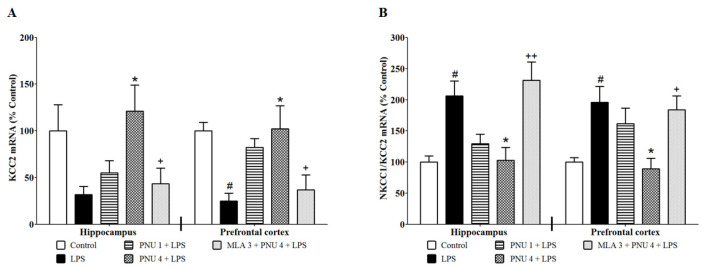
(**A**) Effects of PNU120596 (PNU) on LPS-induced changes in mRNA expression of KCC2 in the hippocampus and prefrontal cortex of mice. LPS (1 mg/kg) clearly decreased KCC2 expression in the hippocampus and prefrontal cortex compared to control. PNU120596 (4 mg/kg) significantly (*p* < 0.05) blocks the effects of LPS on KCC2 expression in the hippocampus and prefrontal cortex. (**B**) Effects of PNU on LPS-induced increases in the mRNA expression ratio of NKCC1 to KCC2 in the hippocampus and prefrontal cortex of mice. LPS (1 mg/kg) significantly (*p* < 0.05) increased NKCC1/KCC2 expression in the hippocampus and prefrontal cortex compared to control. PNU120596 (4 mg/kg) significantly (*p* < 0.05) decreased these changes in the hippocampus and prefrontal cortex. ^#^ *p* < 0.05, LPS (1 mg/kg) vs. control; * *p* < 0.05, PNU (4 mg/kg) plus LPS vs. LPS alone; **^+^** *p* < 0.05 or **^++^** *p* < 0.01, MLA (3 mg/kg) plus PNU (4 mg/kg) plus LPS vs. PNU (4 mg/kg) plus LPS. Data are expressed as the mean ± S.E.M. of *n* = 4–6 mice/group.

**Figure 4 brainsci-14-00290-f004:**
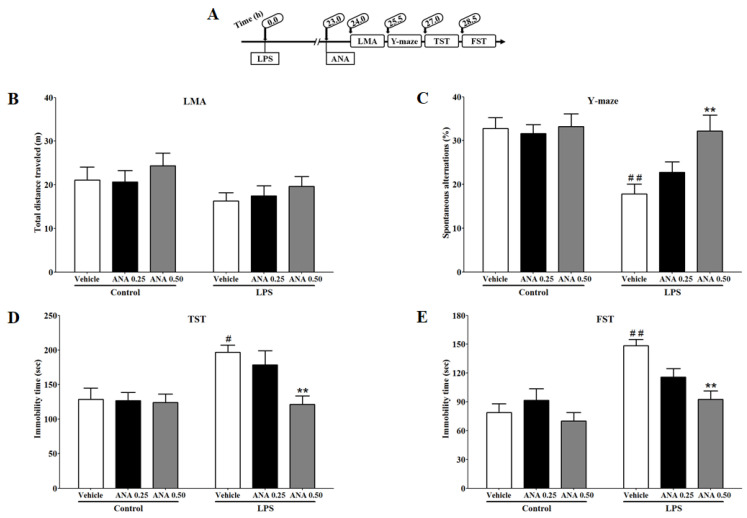
Assessment of antidepressant-like effects of ANA12 (ANA, 0.25 or 0.50 mg/kg) in an inflmmatory mouse model of depression: (**A**) Experimental timeline for drug administrations and behavioral tests (**B**) Effects of ANA on the total distance traveled (m) in LMA. The treatments did not show significant effects on total distance traveled. (**C**) Effects of ANA on spontaneous alternations (%) in Y-maze. ANA (0.50 mg/kg) significantly (*p* < 0.01) increased spontaneous alternations compared to LPS-treated mice. (**D**) Effects of ANA on immobility time (s) in TST. ANA (0.50 mg/kg) significantly (*p* < 0.01) prevented LPS-induced increases in immobility time. (**E**) Effects of ANA on immobility time (s) in FST. ANA (0.50 mg/kg) significantly (*p* < 0.01) prevented LPS-induced increases in immobility time. ^#^ *p* < 0.05 or ^##^ *p* < 0.01, vehicle/LPS (1 mg/kg) vs. vehicle/control; ** *p* < 0.01, ANA (0.50 mg/kg) plus LPS vs. vehicle/LPS. Data are expressed as the mean ± S.E.M. of *n* = 6–7 mice/group.

**Figure 5 brainsci-14-00290-f005:**
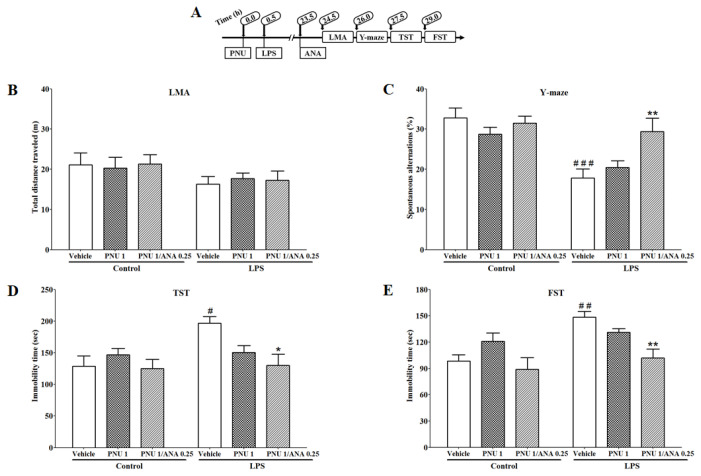
Assessment of enhancing antidepressant-like effects of PNU120596 (PNU, 1 mg/kg) by coadministration with ANA12 (ANA, 0.25 mg/kg) in mice: (**A**) Experimental timeline for drug administrations and behavioral tests. (**B**) Combination effects of PNU and ANA on total distance traveled (m) in LMA. The treatments did not show significant effects on the total distance traveled. (**C**) Combination effects of PNU and ANA on spontaneous alternations (%) in Y-maze. PNU and ANA coadministration significantly (*p* < 0.01) increased spontaneous alternations compared to LPS-treated mice. (**D**) Combination effects of PNU and ANA on immobility time (s) in TST. PNU and ANA coadministration significantly (*p* < 0.05) decreased immobility time compared to LPS-treated mice. (**E**) Combination effects of PNU and ANA on immobility time (s) in FST. PNU and ANA coadministration significantly (*p* < 0.01) decreased immobility time compared to LPS treated mice. ^#^ *p* < 0.05, ^##^ *p* < 0.01 or ^###^ *p* < 0.001, vehicle/LPS (1 mg/kg) vs. vehicle/control; * *p* < 0.05 or ** *p* < 0.01, PNU (1 mg/kg) plus ANA (0.25 mg/kg)/LPS vs. vehicle/LPS. Data are expressed as the mean ± S.E.M. of *n* = 6–10 mice/group.

**Figure 6 brainsci-14-00290-f006:**
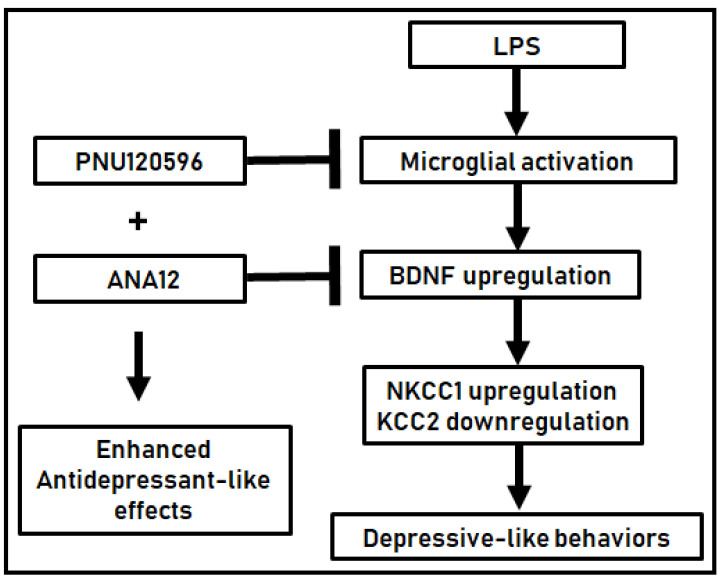
A schematic showing the interaction between α7 nAChR and the TrkB receptor associated with PNU120596’s antidepressant-like effects in an inflammatory mouse model of depression. PNU120596 stimulates α7 nAChR through positive allosteric modulation. Receptor activation prevents an LPS-induced increase in BDNF and NKCC1 expression and KCC2 expression in the hippocampus (HPC) and prefrontal cortex (PFC). Antagonism of TrkB, the receptor for BDNF, using ANA12 decreases LPS-induced depression-like behavior. ANA12 enhances the antidepressant-like effects of PNU120596, indicating the interaction between α7 nAChR and TrkB mediating LPS-induced depression-like behavior. →: stimulation; ─┤: inhibition; +: synergism.

**Table 1 brainsci-14-00290-t001:** Sequence of primers used in the current investigation in qRT-PCR.

Gene	Primer Sequence (5′-3′)
KCC2	AGCCTATGACGATGACCCA (forward)
	CCACCTCTGCTGTCTACATC (reverse)
NKCC1	GGTATCATTAACATTGCCAGTGG (forward)
	CAGATCCTCAGTCAGCCATAC (reverse)
GAPDH	GTGGAGTCATACTGGAACATGTAG (forward)
	AATGGTGAAGGTCGGTGTG (reverse)

## Data Availability

All data included in this study are available upon request from the corresponding author. The data are not publicly available due to institutional copyright policy.

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
