# Peer review of "The α-7 Nicotinic Receptor Positive Allosteric Modulator Alleviates Lipopolysaccharide Induced Depressive-like Behavior by Regulating Microglial Function, Trophic Factor, and Chloride Transporters in Mice"

_brainsci, 2024, doi:10.3390/brainsci14030290_

Round 1
Reviewer 1 Report
Comments and Suggestions for Authors
The present study explores the antidepressive effect of PNU120596, a positive allosteric modulator of α7 nAChR, in the LPS-induced model of depression, along with changes in the expression of BDNF and chloride transporters. The results are interesting and well-presented. However, they should be interpreted with more caution in discussion.
Specific comments:
1. The authors need to clearly state that their results regarding the BDNF increase in the hippocampus and prefrontal cortex are in contrast to the majority of existing literature and should avoid the conclusion that such increase is related to the development of depressive-like behavior. Neurotrophic effects of BDNF in these brain regions are considered neuroprotective. Namely, both depressed patients and animal models of depression, including neuroinflammatory models, show a decrease of BDNF signalling in cortical and hippocampal regions, while antidepressant effects normalize this reduction (Duman et al., 2021; Guan and Fang, 2006). On the other hand, the increase of BDNF signalling in the mesolimbic pathway is related to pro-depressive effects, while its attenuation shows antidepressive properties (Duman et al., 2021; Berton et al., ref 50 and Eisch et al., ref 54). Some of these references are mentioned in the discussion, but the authors should clearly state that the pro-depressive effect of BDNF refers to the mesolimbic pathway, not the hippocampus and prefrontal cortex (Berton et al., ref 50 and Eisch et al., ref 54). Furthermore, studies showed that the antidepressant effect of ANA12, a TrkB inhibitor, is specifically related to its effects on the mesolimbic pathway (Zhang et al., 2014). The authors should refer to all these contradictory findings and discuss their results in light of these discrepancies, considering the methodological differences between studies.
References:
· Duman, et al. Role of BDNF in the pathophysiology and treatment of depression: Activity-dependent effects distinguish rapid-acting antidepressants. Eur J Neurosci. 2021 Jan;53(1):126-139.
· Guan and Fang. Peripheral immune activation by lipopolysaccharide decreases neurotrophins in the cortex and hippocampus in rats. Brain, Behavior, and Immunity, 2006, 20 (1).
· [50] Berton et al. Essential role of BDNF in the mesolimbic dopamine pathway in social defeat stress. Science 2006, 311, 864-868.
· [54] Eisch et al. Brain-derived neurotrophic factor in the ventral midbrain-nucleus accumbens pathway: a role in depression. Biol Psychiatry 2003, 54, 994-1005.
· Zhang et al. Antidepressant effects of TrkB ligands on depression-like behavior and dendritic changes in mice after inflammation. Int J Neuropsychopharmacol. 2014, 18(4): pyu077.
2. In the introduction, the authors should more clearly state their rationale for hypothesizing that the activation of α7 nAChR through BDNF, NKCC1, and KCC2 would cause an antidepressant effect.
3. The authors should specify which part of the prefrontal cortex is analysed by immunofluorescence, and presented in Figure 2. Also, which parts of the hippocampus and prefrontal cortex are dissected and analysed by Western blot? Are the bands on Western blot representing individual animals or pooled samples?
4. Since each experiment has three treatments, did each animal that did not receive all three treatments receive i.p. injections of the appropriate vehicle, e.g., control animals received three i.p. injections of vehicle solutions in appropriate time points?
Author Response
Manuscript No. brainsci-2915909
Response to Reviewer’s comments
We appreciate comments and suggestions raised by the reviewer. Accordingly, we have revised the manuscript. Below is our response:
Reviewer # 1
Comment 1: The authors need to clearly state that their results regarding the BDNF increase in the hippocampus and prefrontal cortex are in contrast to the majority of existing literature and should avoid the conclusion that such increase is related to the development of depressive-like behavior. Neurotrophic effects of BDNF in these brain regions are considered neuroprotective. Namely, both depressed patients and animal models of depression, including neuroinflammatory models, show a decrease of BDNF signalling in cortical and hippocampal regions, while antidepressant effects normalize this reduction (Duman et al., 2021; Guan and Fang, 2006). On the other hand, the increase of BDNF signalling in the mesolimbic pathway is related to pro-depressive effects, while its attenuation shows antidepressive properties (Duman et al., 2021; Berton et al., ref 50 and Eisch et al., ref 54). Some of these references are mentioned in the discussion, but the authors should clearly state that the pro-depressive effect of BDNF refers to the mesolimbic pathway, not the hippocampus and prefrontal cortex (Berton et al., ref 50 and Eisch et al., ref 54). Furthermore, studies showed that the antidepressant effect of ANA12, a TrkB inhibitor, is specifically related to its effects on the mesolimbic pathway (Zhang et al., 2014). The authors should refer to all these contradictory findings and discuss their results in light of these discrepancies, considering the methodological differences between studies.
References:
- Duman, et al. Role of BDNF in the pathophysiology and treatment of depression: Activity-dependent effects distinguish rapid-acting antidepressants. Eur J Neurosci. 2021 Jan;53(1):126-139.
- Guan and Fang. Peripheral immune activation by lipopolysaccharide decreases neurotrophins in the cortex and hippocampus in rats. Brain, Behavior, and Immunity, 2006, 20 (1).
- [50] Berton et al. Essential role of BDNF in the mesolimbic dopamine pathway in social defeat stress. Science 2006, 311, 864-868.
- [54] Eisch et al. Brain-derived neurotrophic factor in the ventral midbrain-nucleus accumbens pathway: a role in depression. Biol Psychiatry 2003, 54, 994-1005.
- Zhang et al. Antidepressant effects of TrkB ligands on depression-like behavior and dendritic changes in mice after inflammation. Int J Neuropsychopharmacol. 2014, 18(4): pyu077.
Our Response: We appreciate the thoughtful comments raised by the reviewer. We have included the following information in the revised version.
Indeed, BDNF has been implicated in the modulation of depressive-like behavior in rodents. “Along these lines, though somewhat controversial, it is noteworthy to mention that BDNF in the ventral tegmental area-nucleus accumbens (VTA-NAc) pathway is involved in the development of depression-like behavior [50,54].” For example, mice experiencing social defeat stress exhibit depressive-like behavior associated with central BDNF upregulation in the mesolimbic dopamine pathway [50,78]. Consistent with a previous report, upregulation of BDNF in the hippocampus is associated with LPS-induced depressive-like behavior [51]. Mice reared in communal nests show upregulation of hippocampal BDNF expression and depressive-like behavior [52]. Mice genetically modified to overexpress BDNF show anxiety-like behavior when exposed to chronic immobilization stress [53]. In an inflammatory mouse model of depression, anxiety-like behavior is comorbid with depressive-like behavior [15] as an indication for involving BDNF in depressive-like behavior. However, viral-mediated suppression of the BDNF receptor TrkB in the VTA and NAc exhibits antidepressant-like behavior [54]. BDNF knockout mice exhibit antidepressant-like behavior in the FST [21], suggesting the regulatory function of BDNF in depressive-like behavior. Antidepressants such as imipramine and fluoxetine were reported to reduce BNDF expression in the rodent hippocampus [51,55]. “Based on all the aforementioned studies, it is important to note that BDNF expression is distinct in specific brain regions. BDNF generally appears to have antidepressant effects in the hippocampus and prefrontal cortex [78,79]. In contrast, BDNF exhibits pro-depressive effects in the VTA-NAc pathway [50,54,78]. Thus, BDNF overexpression in distinct brain regions has been attributed to a high degree of variability of stress models, microglial activators, and the time of measuring these trophic responses.”
Previous studies have shown that ANA12 reduces depressive-like behavior in mice. The antidepressant-like effects of ANA12 have been suggested to be due to blockade of TrkB receptors in the NAc [73,74].
- Duman, R.S.; Deyama, S.; Fogaça, M.V. Role of BDNF in the pathophysiology and treatment of depression: Activity‐dependent effects distinguish rapid‐acting antidepressants. European Journal of Neuroscience 2021, 53, 126-139.
- Guan, Z.; Fang, J. Peripheral immune activation by lipopolysaccharide decreases neurotrophins in the cortex and hippocampus in rats. Brain, behavior, and immunity 2006, 20, 64-71.
Comment 2: In the introduction, the authors should more clearly state their rationale for hypothesizing that the activation of α7 nAChR through BDNF, NKCC1, and KCC2 would cause an antidepressant effect.
Our Response: We appreciate the comment. Accordingly, we have added the following information in the revised version as shown below.
The α7 nAChRs are included in the modulation of microglial activation, the primary source of neuroinflammation in the brain [37-39]. Moreover, these receptors are widely expressed in MDD-relevant brain regions, such as the hippocampus and prefrontal cortex, implicated in the regulation of emotion and behavior [35,40,41]. Therefore, the anti-inflammatory role of α7 nAChRs could be a promising target to treat disorders associated with inflammation [36,42]. “Microglial activation causes the release of BDNF as an inflammatory mediator and is involved in the dysregulation of chloride ion concentration which is regulated by GABAergic neurotransmission via NKCC1 upregulation and KCC2 downregulation. Interruption of intracellular chloride ion concentration subsequently induces neuronal excitability. Thus, targeting BDNF, NKCC1 and KCC2 within the hippocampus and prefrontal cortex via α7 nAChR PAM might have potential therapeutic utility for MDD.”
Comment 3: The authors should specify which part of the prefrontal cortex is analysed by immunofluorescence and presented in Figure 2. Also, which parts of the hippocampus and prefrontal cortex are dissected and analysed by Western blot? Are the bands on Western blot representing individual animals or pooled samples?
Our Response: We have specified the prefrontal cortex region in the revised version. For the Western blot analysis, DG and CA1 region of the hippocampus and medial part of prefrontal cortex were dissected as shown below. The bands are representing individual animals.
Figure 2. (A) Effects of PNU120596 (PNU) on immunoreactivity of BDNF in DG, CA1 and medial region of prefrontal cortex of mice.
Comment 4: Since each experiment has three treatments, did each animal that did not receive all three treatments receive i.p. injections of the appropriate vehicle, e.g., control animals received three i.p. injections of vehicle solutions in appropriate time points?
Our Response: Yes, control animals received three i.p. injections of vehicle solutions in appropriate time points.
Reviewer 2 Report
Comments and Suggestions for Authors
This study used a mice model of major depressive disorder (MDD) induced by LPS and primarily explored the role of a positive allosteric modulator (PAM) of the α7 nAChR, namely PNU120596, on the expression of BDNF, NKCC1, and KCC2 in the cortex and hippocampus of animals. The results are consistent and indicate that the positive modulation of the α7nAChR may alleviate neuroinflammation caused by LPS and that the combination of PNU120596 and ANA12 may prevent depressive-like behavior in mice. MDD is a debilitating disorder that affects the quality of life of patients. So, the findings presented by the authors could provide new insights for the treatment of the underlying mechanisms of this disease. Nevertheless, I have spotted some issues that should be addressed by the authors before acceptance.
1) The abstract should inform that the study was performed using male C57BL/6J mice.
2) Section 2.1. Animals: Please provide the weight of the animals used for the protocols.
3) Section 2.6. Quantitative real-time polymerase chain reaction: Please inform the sample and reagent volumes used for the analysis.
4) Section 2.7. Behavioral tests: I suggest the authors separate the description of the behavioral tests into subsections: Y-maze, TST, and so on.
5) The quality of Fig. 2 (Immunofluorescence) should be improved.
6) The authors have significant results from different protocols. So, I suggest them to improve Fig. 6 to better depict the findings of the study.
7) MDD has an important neuroinflammatory component. So, what about investigating some inflammatory cytokines, such as interleukin-18 (IL-18), which has been cited here? Since the α7 nAChRs present an anti-inflammatory role investigating some inflammatory markers would be of great aid to this or future studies.
8) Just a curiosity: Did the animals show any responses, such as fever, regarding the LPS administration?
9) Please correct some typing and punctuation errors throughout the text.
Author Response
Manuscript No. brainsci-2915909
Response to Reviewer’s comments
We appreciate comments and suggestions raised by the reviewer. Accordingly, we have revised the manuscript. Below is our response:
Reviewer # 2
Comment 1: The abstract should inform that the study was performed using male C57BL/6J mice.
Our Response: We have included the mouse strain in the revised version as shown below.
“Experiments were conducted in male C57BL/6J mice.”
Comment 2: Section 2.1. Animals: Please provide the weight of the animals used for the protocols.
Our Response: We have included the mouse weight in the revised version as shown below.
“Male C57BL/6J mice (weighing 20-30 g) were purchased from Jackson Laboratory (Bar Harbor, ME, USA).”
Comment 3: Section 2.6. Quantitative real-time polymerase chain reaction: Please inform the sample and reagent volumes used for the analysis.
Our Response: We have provided the sample and reagent volumes in the revised version as shown below.
Total RNA was isolated from brain tissue using TRIzol reagent (Invitrogen, Carlsbad, CA, USA) according to manufacturer’s instructions. “One microgram of total RNA
was reverse transcribed using a reverse transcription system to make
cDNA using High-Capacity cDNA Reverse Transcription Kit” (Applied Biosystems, Carlsbad, CA, USA) and Master Cycler Personal (Eppendorf, Hauooauge, NY, USA) as described previously [38]. “Approximately 20 µl of cDNA was yielded, out of which 1 μl was used for each real-time PCR analysis.” Primers sequences were obtained from Integrated DNA Technologies (Coralville, Iowa, USA) and are displayed in Table 1. The Ct value was measured for each gene and relative expression of each gene was calculated using delta-delta Ct method.
Comment 4: Section 2.7. Behavioral tests: I suggest the authors separate the description of the behavioral tests into subsections: Y-maze, TST, and so on.
Our Response: We have made the subsections in the revised version as shown below.
2.7.1. Locomotor activity:
2.7.2. Y-maze
2.7.2. Tail suspension test
2.7.4. Forced swim test
Comment 5: The quality of Fig. 2 (Immunofluorescence) should be improved.
Our Response: We have replaced with new images.
Comment 6: The authors have significant results from different protocols. So, I suggest them improve Fig. 6 to better depict the findings of the study.
Our Response: We have improved the figure in the revised version for better clarity.
Comment 7: MDD has an important neuroinflammatory component. So, what about investigating some inflammatory cytokines, such as interleukin-18 (IL-18), which has been cited here? Since the α7 nAChRs present an anti-inflammatory role investigating some inflammatory markers would be of great aid to this or future studies.
Our Response: We appreciate the comment. In future studies, we will determine the effects of α7 nAChRs PAM on inflammatory cytokines such as IL-18.
Comment 8: Just a curiosity: Did the animals show any responses, such as fever, regarding the LPS administration?
Our Response: We did not observe any such responses.
Comment 9: Please correct some typing and punctuation errors throughout the text.
Our Response: We have made all the corrections in the revised version.